# Advancements and Prospects in Perovskite Solar Cells: From Hybrid to All-Inorganic Materials

**DOI:** 10.3390/nano14040332

**Published:** 2024-02-08

**Authors:** Fernando Velcic Maziviero, Dulce M. A. Melo, Rodolfo L. B. A. Medeiros, Ângelo A. S. Oliveira, Heloísa P. Macedo, Renata M. Braga, Edisson Morgado

**Affiliations:** 1Postgraduate Program in Chemistry, Federal University of Rio Grande do Norte, Natal 59078-970, Brazil; fernando.velcic.017@ufrn.edu.br; 2Laboratório de Tecnologia Ambiental—LABTAM, Federal University of Rio Grande do Norte, Natal 59078-970, Brazil; rodolfo.luiz.medeiros@ufrn.br (R.L.B.A.M.); angelo_quimica@hotmail.com (Â.A.S.O.); helo.pimenta@hotmail.com (H.P.M.); renata.braga@ufrn.br (R.M.B.); 3Postgraduate Program in Materials Science and Engineering, Federal University of Rio Grande do Norte, Natal 59078-970, Brazil; 4Agricultural School of Jundiaí, Federal University of Rio Grande do Norte, Macaíba 59280-000, Brazil; 5Postgraduate Program in Chemical Engineering, Federal University of Rio Grande do Norte, Natal 59078-970, Brazil; 6PETROBRAS R&D Centre (CENPES), Rio de Janeiro 21941-915, Brazil; emorgado@petrobras.com.br

**Keywords:** solar cells, perovskites, hybrid, inorganic, lead-free

## Abstract

Hybrid perovskites, materials composed of metals and organic substances in their structure, have emerged as potential materials for the new generation of photovoltaic cells due to a unique combination of optical, excitonic and electrical properties. Inspired by sensitization techniques on TiO_2_ substrates (DSSC), CH_3_NH_3_PbBr_3_ and CH_3_NH_3_PbI_3_ perovskites were studied as a light-absorbing layer as well as an electron–hole pair generator. Photovoltaic cells based on per-ovskites have electron and hole transport layers (ETL and HTL, respectively), separated by an ac-tive layer composed of perovskite itself. Major advances subsequently came in the preparation methods of these devices and the development of different architectures, which resulted in an efficiency exceeding 23% in less than 10 years. Problems with stability are the main barrier to the large-scale production of hybrid perovskites. Partially or fully inorganic perovskites appear promising to circumvent the instability problem, among which the black perovskite phase CsPbI_3_ (α-CsPbI_3_) can be highlighted. In more advanced studies, a partial or total substitution of Pb by Ge, Sn, Sb, Bi, Cu or Ti is proposed to mitigate potential toxicity problems and maintain device efficiency.

## 1. Introduction

The new generation of photovoltaic cells, also called emerging technologies or third-generation cells, has been highlighted in the scientific community and companies in the last ten years, gathering great efforts to increase efficiency, reproducibility and stability in order to make their commercialization possible to meet increasingly demanding markets [1]. Promising materials include perovskites due to their unique proposal to combine low cost with high efficiency when applied on a large scale. Perovskites are traditionally prepared via solid-state synthesis at elevated temperatures (usually above 1000 °C). However, in the early 1990s, a group of researchers from IBM and the University of Houston (USA) investigated the optoelectronic properties of hybrid perovskites for transistor and diode applications [2]. In the study, Mitzi and co-workers [2] showed an increase in the binding energy of the excitons (quasiparticle) from the combination of three-dimensional layers of ABX_3_ (R-NH_3_PbI_3_) perovskites organized under organic structure (R-NH_3_)_2_PbI_4_ planes. However, according to the review article by Snaith [3], the focus on researching layered “excitonic materials” may have been responsible for the lack of research development of these materials for solar energy, as the layered organization of perovskites is not effective for such an application. In this aspect, the structural conformation, which influences the optical properties, is the key to obtain solar devices with higher efficiency.

Sensitized systems were first proposed with organic components as a pathway for developing alternative materials [4]. However, these materials are limited in their light conversion capacity due to their low absorption coefficients and narrow bands. Thus, researchers have been studying quantum dots (QD) such as PbS, InAs, CdS, CdSe and InP in order to circumvent this issue [5,6,7,8,9]. Yet, despite the intense light absorption by the band gap of the quantum dots, significant losses in light utilization and charge separation are observed in the semiconductor–sensitizer interface. Therefore, improved structures and properties are needed to give the system enhanced ability to absorb light and generate current. Organic–inorganic hybrid perovskites appeared to exhibit a unique combination of optical, excitonic and electrical conductivity properties. In the literature, the working principle of perovskites as solar cells has not yet been completely unraveled. Simply put, the photovoltaic process starts with the absorption of light, resulting in a quasi-Fermi level generated by the splitting of electrons and holes. The second step consists of charge separation, using selective contacts for each quasi Fermi level [10].The first hybrid perovskites were lead-based sensitized perovskites containing halide groups (CH_3_NH_3_PbBr_3_ and CH_3_NH_3_PbI_3_) and n-type semiconductors such as TiO_2_ (dye-sensitized solar cell—DSSC) [11], which showed an impressive efficiency of 3.8% for the time. Currently, a series of other organic cations containing NH_x_^+^ groups have been proposed as components in hybrid perovskites, such as formamidine (FA), butyl ammonium (BA), cyclopropylamine (CA), and naphthalene methylammonium (NMA) [11,12,13]. Fourteen years later, the efficiency already exceeds 25%, matching that of commercial silicon-based solar cells [14], as a result of great efforts in investigating the structure and properties of the materials, controlling crystal growth and the morphology of the perovskite layer, developing film deposition techniques, modifying cell architecture and using different anti-solvents and dopants [15,16,17,18,19,20]. This review work has as its main objective to discuss the advances of perovskites applied in solar cells, highlighting the main points necessary to obtain completely inorganic, stable and lead-free structures.

## 2. Hybrid Perovskite Structure and Its Components

It is essential to understand the former technologies to better understand the functioning of perovskite-based photovoltaic cells. A traditional solar cell is basically made up of two semiconductor (junction) layers: one layer containing a positive semiconductor (p-type) and another containing a negative semiconductor (n-type), which generates an electric current when exposed to solar radiation [21]. Solar cells are classified in generations, being differentiated by the materials and processing technologies employed in their manufacture. The first generation, also known as wafer cells (Figure 1) [22], are based on the p–n junction and include the crystalline silicon solar cells, which are the most commercialized technology on the market today, with efficiencies between 15 and 20%. However, even despite the sharp fall in prices in recent years, they still have a high production and installation cost [22]. 

Second-generation or commercial thin-film solar cells, such as amorphous silicon-based films, CIGS (copper indium gallium selenide), or CdTe (cadmium telluride) thin films, have lower manufacturing costs compared to those of crystalline silicon, although they still require vacuum processes and heat treatments at high temperatures. However, second-generation modules in general have lower efficiency than that of the previous ones. Third-generation or emerging thin film cells represent the most recent technology in the scientific world today [23]. These are based on organic materials, dyes, quantum dots, or perovskites, and encompass more complex structures such as tandem (or multi-junction) cells. These cells generate multiple excitons, offering the possibility of exceeding the theoretical efficiency limits of Shockley and Queisser (S-Q) in addition to associating high efficiency and low cost [23,24,25].

The last decade has seen an unprecedented and rapid increase in perovskite-based solar cells [3,26,27,28,29]. The focus of several research groups at present lies in utilizing these materials for photovoltaic devices, showing the considerable evolution of the components (HTL, active layer, ETL, and electrical contacts) in such devices and of its efficiency (power conversion efficiency—PCE), which is currently certified at 25.5% according to the US National Renewable Energy Laboratory (NREL) [28].

**Figure 1 nanomaterials-14-00332-f001:**
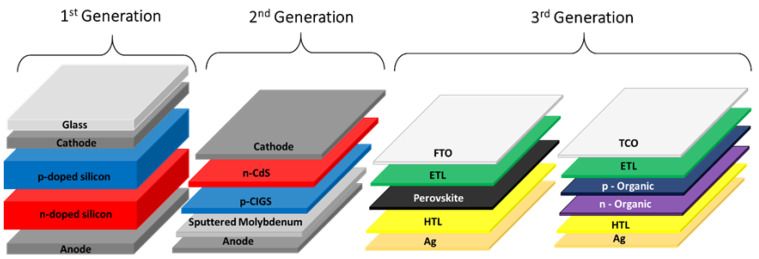
Representation of the solar cells generations: 1st generation, 2nd generation (commercial thin films), and 3rd generation (emerging thin films). Adapted from [30,31].

In addition to their high efficiency, perovskite solar cells (PSCs) can be prepared with materials and techniques that are potentially cost-effective and enable large-scale commercialization [30,31]. The unique properties of these absorbent materials provide numerous advantages in optoelectronic applications, which in many aspects originate from the nature of the perovskite structure [32]. 

Hybrid perovskites composed of lead halide and methylammonium have been shown to be promising photovoltaic materials due to their properties, such as their (i) high absorption coefficient in the visible region of the solar spectrum; (ii) direct (due to the alignment of valence and conduction bands) and tunable band gap; (iii) ambipolar transport of positive and negative charges; (iv) high mobility of excitons compared to that of those in organic semiconductors; and (v) diffusion length exceeding micrometer scale in single crystals [33,34,35,36].

In a typical PSC, the photoabsorber layer is composed of perovskite sandwiched between an electron transport layer (ETL), which is an n-type semiconductor, and a hole transport layer (HTL), the latter usually SPIRO-OMeTAD or PEDOT: PSS [37,38,39]. One of the layers, HTL or ETL, is deposited on a transparent conductive electrode, either FTO (fluorine-doped tin oxide) or ITO (indium tin oxide), and a metal (Au, Ag or Al) is deposited as the rear contact, completing the device. When ETL (usually TiO_2_ or SnO_2_) is deposited on the glass electrode, PSC is generally called the conventional type. However, when ETL (generally fullerene-based molecules) is deposited on the perovskite layer, the device is named the inverted type.

The operation principle starts with light absorption by perovskite to generate the electron–hole pair. The electrons are conducted via the electron transport layer (ETL), while holes are directed by the HTL (hole transport layer), promoting charge separation so that recombination of the electron–hole pair is avoided. The electrons then flow through the external circuit, producing an electric current. 

Figure 2 shows the energy band diagram of typical PSC stacking FTO/TiO_2_/perovskite/spiro-OMTAD/Au, which exemplifies the operation principle. Although it is the most accepted principle today, the charge generation and transport (electrons and holes) processes are not as clear as they are in amorphous silicon cells. In the device architecture, the layers formed by TiO_2_ and Spiro-OMeTAD act as charge separators. The TiO_2_ layer (ETL) deposited on an FTO-based electrode extracts the electron, while the hole is extracted by Spiro-OMeTAB (the HTL layer) in contact with the gold electrode. These materials are examples of electrodes and electron-carrying layers and holes. Other materials such as ITO, niobium, zinc and nickel oxides, in addition to aluminum, silver, copper and carbon-based contacts are used as alternative components [40,41,42].

Perovskite solar cells can be arranged in regular or inverted configurations. The regular configuration is originated from solid-state DSSCs and is characterized by layers in the fowling order of FTO/ETL/perovskite/HTL/metal. The inverted configuration originates from regular organic solar cells with the ETL and HTL layers in inverted positions, in which ITO is generally the transparent conductive electrode used [43]. Figure 3 summarizes the origin of the different architectures of perovskite-based solar cells.

In addition to the cell architecture, the structural organization of the perovskite layer also imparts different optoelectronic properties to the device, potentially leading to improvements in stability, conversion efficiency and charge transport. This is evident in the case of 2D perovskites with a Ruddlesden–Popper arrangement (2DRP), which differs from traditional 3D perovskites in featuring a general formula of A_n+1_B_n_X_3n+1_ or AX(ABX_3_)_n_ (n = 1, 2, 3, …). These structures consist of alternating layers of face-centered cubic structures (AX) and perovskite layers (ABX_3_) along the crystallographic Z-axis, offering enhanced optoelectronic characteristics [46,47]. This conformation allows for the insertion of long-chain organic cations between the perovskite layers, inducing octahedral distortions, altering Coulombic interactions, and creating trap states and quantum confinement. This results in controlled variations of the band gap and charge carrier recombination rates [14]. In one of the most prominent works in the field, Tsai H. and colleagues asserted that one of the main contributors to the low power conversion efficiency (PCE) achieved in solar cells containing 2DRP-type perovskites is due to the inhibition of charge transport by the organic cations. These cations act as insulating layers spacing out the conductive inorganic plates. As an alternative approach, the authors produced a near-single-crystalline-quality thin film, which exhibited a strongly preferential out-of-plane alignment. This resulted in efficient charge transport, achieving a PCE of 12.52%, along with demonstrating good stability under heat and humidity conditions [48].

Another explored alternative is tandem (multi-junction) cells, which consist of two or more subcells stacked in series or parallel. Currently, this configuration type has been reported to achieve the highest reported efficiency values, surpassing a PCE (power conversion efficiency) of 45% [44]. This configuration is commonly reported using either two terminals (2T) or four terminals (4T) to optimize the spectral range absorbed by the device, and to reduce overall thermalization losses and photon transmission losses. However, the tandem configuration still faces challenges in commercialization due to its higher costs and difficulties in the manufacturing process [45,49].

### Advances in Lead-Based Hybrid Perovskites

Totally inorganic perovskites such as SrTiO_3_, BaTiO_3_ and Pb (Zr, Ti)O_3_ have been extensively explored in the past due to their excellent dielectric, ferroelectric and piezo-electric properties, acting in numerous applications. However, hybrid perovskites (hybrid organic-inorganic perovskites, HOIPs), know more than a century ago, were only recently considered a promising class of materials for applications in solar cells. Due to the remarkable advancement in their photovoltaic performance, hybrid perovskites are currently at the epicenter of global investigations [50]. In general, this type of material has a structural formula of the ABX_3_ type, in which the A site is occupied by monovalent cations and the B site is occupied by divalent cations. The best-known structure is CH_3_NH_3_PbI_3_, which presents an efficiency of above 25% [28]. 

On the other hand, many of these materials are polar substitutes and, therefore, not viable for classical inorganic perovskites. The origin of polar properties in various classes of HOIPs (both single crystals and thin films) has been described by Xu et al., in which they provide a comprehensive explanation of their piezoelectric and pyroelectric responses, and discuss their possible applications as capacitors, memory elements, piezoelectric transducers, and pyroelectric sensors [51]. Several classes of HOIPs exhibit non-symmetrical structures under ambient conditions, and the number of this type of material is constantly growing due to the large number of possibilities in the replacement of functional groups of the perovskite network and the changes in properties generated by these modifications. In their review, Xu et al. [51] analyzed the possible mechanisms for obtaining the polar phase that would result in ferroelectric properties in HOIPs, concluding that this type of material has high potential to be explored in future applications such as memory elements, transducers and piezoelectric generators. The investigation of the polar properties of HOIPs is in constant development, and there is still much to be explored.

In a work published in Coordination Chemistry Reviews [52], it was pointed out that semiconductors based on quantum dots (QDs) and hybrid perovskites emerged as the two main materials with potential for high-performance photovoltaic applications, showing rapid progress in the last decade. In their review, Zhou et al. elaborates an important discussion about the synergistic combination of QDs and perovskites, which results in a series of promising alternatives for future advances in hybrid solar cells, presenting important points such as the promotion of the growth of perovskite crystals, light absorption in a wide spectrum range, fast electron–hole pair separation, structural stability, and improved interface engineering [52]. The greatest emphasis has been placed on the PbS/CH_3_NH_3_PbI_3_ system, which allows the heteroepitaxial growth of perovskites in QDs. Topics such as fundamental physical and chemical properties, structural integrity, material and film processing techniques, and energy transfer between QDs and perovskites, as well as advances in device architecture and photovoltaic performance are under constant focus and development [52].

Regarding thermodynamics, Ivanov et al. studied the enthalpy of the synthesis of hybrid perovskite CH_3_NH_3_PbX_3_ (X = Cl, Br, I) prepared in dimethylsuffoxide (DMSO) solution using the technique of solution calorimetry [53]. The standard enthalpy (∆H^0^) and Gibbs free energy (∆G) of hybrid perovskites CH_3_NH_3_PbX_3_ (X = Cl, Br, I) were obtained experimentally at 298 K and compared with data available in the literature. The calculated results proved to be in agreement with the value found in other works in which the ∆G was evaluated based on results of the measurement of vapor pressure in the system, indicating the formation of solid PbX_2_, HX gas, and methylamine from the decomposition of CH_3_NH_3_PbX_3_. Formation entropy has been shown to play an important role in the stability of hybrid organic and inorganic perovskites with regard to their decomposition into constituent halides [53].

Complementarily, Xue et al. [19] investigated the influence of temperature on the formation of two-dimensional perovskite (PEA)_2_(MA)_2_Pb_3_I_10_ films based on hot casting and post-annealing, developing two-dimensional (2D) layered perovskites, in which the inorganic semiconductor layers act as “wells” while the insulating organic layers act as “barriers”, resulting in 2D perovskites with better moisture resistance compared to that of their 3D analogues. It was also confirmed that the formation of (PEA)_2_(MA)_2_Pb_3_I_10_ crystals via the hot melting route showed better crystallinity when compared to that under post-annealing, in addition to showing that thermal annealing and mass transfer in the solvent are crucial in the two-dimensional perovskite crystallization process. As a result, the devices achieved an energy conversion efficiency of 5.57% at 100 °C (hot casting), optimizing the substrate heating temperature and the two-dimensional hot casting films (PEA)_2_(MA)_2_Pb_3_I_10_ that began to show stability under environmental conditions.

The manufacture, processing and durability of devices employing hybrid perovskite are critically influenced by the active area, stability and cost of the proposed materials, which represents possible drawbacks for the commercial upscaling of PSCs. Ji et al. [54] conducted a review of recent mechanical studies of known HOIPs. They initially presented a comprehensive account of the relationship between crystalline structures and properties, in which the influence of different chemical factors on HOIP load extraction was widely discussed. HOIP functional abnormalities under stress stimuli were also reviewed to address the effects of negative linear compressibility, a negative Poisson ratio and a barocaloric effect (related to pressure and heat effects). Some properties, such as the compliant nature of thin HOIP film and internal grid voltage or external voltage disturbances can reduce the degree of crystallinity [55,56,57] and lead to the formation of defects [58] with unintended consequences for the electronic properties of the devices [56,57,59,60].

Finally, the mechanical characteristics of HOIPs that have ABX_3_ structures diverge from those observed in their oxide counterparts in two main respects: first, the mechanical properties of HOIPs can be tuned using different combinations of organic molecules to achieve a desired configuration; second, the additional degrees of freedom inherent in the crystal structure give HOIPs a remarkable capacity for mechanical adaptability. In addition, the change in barocaloric effects associated with entropy changes during phase transitions in HOIPs presents great potential for application of this material in solid state cooling, when compared to that of the metallic and inorganic materials conventionally used [54]. It was found that the stress–strain state strongly influences the intrinsic flexibility and stability of the HOIP structures. More and more theoretical models are used that simulate HOIP in relevant device shapes; likewise, an increasing number of experimental methods have been developed to understand more deeply how performance loss is related to effective deformation. In general, the exploration of HOIP mechanics is still an under-addressed topic, and the challenges for HOIP applications indicate that its mechanical properties will be an area of growing interest in research involving specialization in multidisciplinary fields [54].

New-generation semiconductors based on halide perovskites with an ABX_3_ structure have been extensively investigated due to their excellent optical and electronic properties, ideal for applications in optoelectronic devices, including photovoltaic cells, light emitting diodes, photodetectors, and lasers [61]. By manipulating morphological dimensionality, low-dimensional halide perovskites, including 2D nanosheets, 1D perovskite nanowires, and 0D perovskite quantum dots, have been extensively advanced to showcase unique properties compared to those of their counterparts, attributed to quantum size effects. In addition to ABX_3_ perovskites, organic–inorganic halide compounds containing octahedral arrangements (BX_6_) as the fundamental building unit of unit cells can also be assembled to have other types of crystallographic structures. From the combination of organic and inorganic precursors, low-dimensional inorganic halogenated hybrids with 2D, quasi-2D, 2D-wave, 1D and 0D molecular structures were prepared and studied [62,63,64,65]. Due to the effects of quantum confinement and site isolation, these nanoscale molecular metal halide hybrids exhibit discrete values that result in remarkable and unique properties when compared to those of ABX_3_ perovskites.

Due to the growing development in the area of perovskites and nanoscale metal halide hybrids, it is essential to review the recent progress in these areas. Furthermore, there is a need to clarify the difference between low-dimensional metal halide perovskites and molecular metal halide hybrids. In their review article, Zhou et al. [66] discuss the synthesis, characterization, application, and computational studies of low-dimensional metal halide perovskites and hybrids, noting that the development of new organic metal halide hybrids still mainly depends on the trial-and-error-based empirical method, although some researchers can manipulate perovskites with size and shape control to some degree. 

Although it is possible to explain some of the photophysical and electronic properties of organic metal halide hybrids through established theories and computational studies, the kinetics and thermodynamics of these materials, especially in the excited state, remain without detailed understanding. The way in which the organic and inorganic parts interact affects the ground and excited state properties of the topology of the metal halide structure, so these aspects need to be better understood. 

Despite the great interest in studies and the growing number of publications with these materials, organic metal halide hybrids still have low stability when compared to inorganic or even organic semiconductors to compete commercially. Therefore, the detailed study of the factors that cause the degradation of these materials, mainly with regard to humidity, oxidation, and heat, is essential for the development of new materials with greater stability. In addition, there is still a need for a further investigation of the in-fluence of other factors, such as pressure, magnetic field, or polarized light, since this type of study is still scarce in the literature. In addition to what has already been mentioned, other potential applications, such as solar concentrators, magnetic storage, light guides, scintillators, and so on can be promising directions for future research for these materials.

In a recent short communication, the researchers from the Shanghai-China School of Materials Science and Engineering used the one pot method for the controlled synthesis of organic–inorganic hybrid perovskite crystals [67]. A one pot solvothermal process (see Figure 4) for synthesizing single sheet-shaped CH_3_NH_3_PbBr_3_ crystals of a size of 100 µm and a thickness between 3 and 8µm was reported. In addition, the controlled behavior of crystalline surface corrosion, which could be irregular surface collapse caused by the local accumulation of methylammonium cations, is directly related to the Pb^2+^/MA ratio [67].

In a review article published in Materials Today Energy, Torabi et al. [18] pointed out that hybrid perovskite solar cells, as a photoactive layer, have been the subject of a lot of focus in recent years due to their high energy conversion efficiency and potential to produce low-cost photovoltaic modules. Organic–inorganic perovskite solar cells have a record energy conversion efficiency of 23.3% for materials produced at the laboratory scale (Figure 5), which already surpasses that of commercially dominant polycrystalline silicon and CdTe solar cells. Due to factors such as high open circuit voltage, strong absorption edge, adjustable band gap, and solution processability, perovskite solar cells are ideal materials for incorporation into multijunction solar cells, which in practice can exceed even theoretical limits of a single-junction cell. This review addressed key points in the development of single-junction perovskite solar cells, focusing on discussions of material structure, band gap alteration, and crystallization methods, following the theoretical efficiency limits of single- and multiple-junction solar cells. Thus, the challenges of perovskite-based multijunction solar cells were presented, highlighting advances made in recent years with organic and inorganic perovskites and perovskite-perovskite multijunction cells [18].

One of the recent works in the literature addressing the use of lead-based perovskites is that of Meng et al. [17], which aimed to improve the quality of the photoabsorbing perovskite layer and, consequently, obtain high photovoltaic conversion performance from perovskite solar cells (PSCs). In this work, formic acid was used as an additive to the precursor solution of formadinium lead iodide (FAPbI_3_) to reduce the colloidal size in the solution, leading to a more uniform deposition of the FAPbI_3_ film with a lower trapping density and greater mobility of carriers. As a result, the cells that used formic acid in their formulation showed significantly better photovoltaic performance than the FAPbI_3_ reference film without the additive. It was observed that colloidal size is an important parameter that is directly related to the performance of the photovoltaic cell. Within the range studied from 6.7 to 1.0 nm, an inverse proportionality relationship was found in which, the smaller the colloidal size, the greater the efficiency of the solar cell. More specifically, the cell efficiency increased from 17.82% (for the control cell without formic acid) to 19.81% when 0.764 M formic acid was added. Formic acid was also added to a precursor solution of CH_3_NH_3_PbI_3_ (MAPbI_3_), obtaining similar results, in which the presence of the additive generated a 1% increase in the efficiency of the device, which went from 16.07% to 17.00%. Formic acid and other organic acids are effective candidates for tuning the properties of perovskite films to achieve even greater performance. 

In addition to methylammonium, other organic cations have been evaluated as components of the A-site in lead-based hybrid perovskites. Lu J. et al. tested the partial substitution of methylammonium with ethane-1,2-diammonium (EDA^2+^), propane-1,3-diammonium (PDA^2+^), and hexane-1,6-diammonium (HDA^2+^) to assess the influence of an increasing carbon chain length and cation charge on the hybrid organic–inorganic perovskite (HOIP). They reported that the best results were obtained with the addition of 0.8 mol% of EDA^2+^ at the A-site of MAPbI_3_, resulting in an increase in PCE from 17% in the EDA-free perovskite solar cell to 18.6%, along with a significant improvement in cell stability under simulated working conditions (AM 1.5G irradiation, 50% RH, and a 50 °C device temperature). The cell with the EDA additive maintained 75% of its initial efficiency after 72 h of testing, compared to the complete degradation of that within 15 h in the EDA-free cells [68].

Furthermore, Zhang J. et al., through computational studies, demonstrated that the partial substitution of the A-site in CsPbI_3_ perovskites with organic dopants such as dimethylammonium (DMA^+^), ethylammonium (EA^+^), and guanidinium (GA^+^) substantially improves the stability of the α conformation (cubic phase), which exhibits better performance for photovoltaic applications. This improvement is achieved by reducing the volume of the doped cell by 25% using organic cations, resulting in the tilting of the octahedra that compose the structure and a reduction in the extra space at the A-site, facilitating the packing effect in the structure [69].

## 3. Fully Inorganic Perovskites

Perovskites with hybrid structures (organic–inorganic) quickly achieved 23% efficiency from structural modifications and preparation methods. However, corrosion problems of halogen elements and the instability of the organic group as a function of temperature and especially humidity are the main challenges to increasing the production scale, as they require more complex structures and processes [70,71,72,73]. Despite numerous efforts such as passivation treatment and structural modification to increase the stability of hybrid perovskites, problems limiting the performance of these materials have led to the development of structures free of organic groups [63,64,65,66,67,68]. In this sense, optoelectronic properties of all-inorganic perovskites were studied based on the understanding of hybrid perovskite properties. It was initially assumed that the organic cation of hybrid perovskites was responsible for obtaining a long diffusion length, which is the average length a charge moves between generation and recombination. On the other hand, in studies by Stoumpos et al. [74] using X-ray and γ detectors to estimate the electronic mobility of CsPbBr_3_ perovskites, they concluded that a mobility of ~1000 cm^2^V^−1^s^−1^ indicates that the inorganic cation may also have compatible diffusion lengths for application in photovoltaic devices. Thus, CsPbX_3_ (X = Br, I) perovskites have received remarkable attention from the scientific community because of their greater stability compared to that of hybrid perovskites. 

CsPbX_3_-based perovskites (X = F, Cl, Br and I) are semiconductors with orthorhombic, tetragonal and cubic structures. Structural transformation depends on heat treatment; for example, the structure of CsPbBr_3_ is cubic at temperatures above 130 °C [65]. The nanocrystals associated with each crystalline phase directly influence the optoelectronic properties of perovskites by introducing different halide group ions and/or controlling particle size. For this latter purpose, some additives are used, such as oleic acid, octadecene and oleylamine [70]. 

Although all-inorganic perovskites still show lower efficiency than that of hybrids, advances in research already represent an increase in PCE from 2.9% to 10% using the tetragonal phase of CsPb_2_Br_5_ perovskite [70,71,72,73,74,75]. Wang P. et al. [76] reported a 15% PCE using the CsPbI_3_ structure. Therefore, all-inorganic perovskites are receiving attention as a potential material to overcome the hybrid organic group instability problem and maximize thermal stability. One of the challenges in all-inorganic CsPbI_3_-based perovskites is to stabilize the photoactive phase at room temperature, which is the black phase (α-CsPbI_3_) formed at temperatures of approximately 310 °C [77]. Figure 6 illustrates the transformation of the yellow phase, which has an orthorhombic structure (δ-CsPbI_3_), to the black phase (α-CsPbI_3_), which has a cubic structure [77]. The tolerance factor for CsPbI_3_ perovskites is 0.8, which favors the formation of the yellow phase at room temperature, due to the small size of Cs^+^ cations (r = 181 pm), to support the PbI_6_ octahedra in the cubic structure, causing degradation at room temperature (α-CsPbI_3_→δ-CsPbI_3_) [78].

One of the strategies to stabilize the black phase (α-CsPbI_3_) is to increase the structure tolerance factor, which may be achieved by partially exchanging Cs^+^ ions (site A) with larger ionic groups such as MA^+^ (r = 217 pm) or FA^+^ (r = 253 pm), or by partially replacing site X ions with Br^−^ (r = 196 pm) or Cl^−^ (r = 181 pm). Another strategy is to control crystal symmetry via size reduction, which is called “size-controlled stabilization” [77,78,79]. In this case, stabilization is made possible via the use of an additive such as the inclusion of Eu (Eu^2+^ or Eu^3+^). The addition of Eu ions can control grain size and microdeformation in crystals, which can be observed via X-ray diffraction [80]. However, the function of additives has not yet been fully elucidated, as there are few reports in the literature when compared with those on hybrid perovskites. What is well directed is the search for black phase stabilization (α-CsPbI_3_) based on structural modifications via either heat treatment, additives or dopants. Figure 7 shows some of the methods addressed by different authors to enhance the performance and stability of fully inorganic perovskite-based solar cells. 

The use of additives to stabilize the black phase during the synthesis stage without the need for high temperatures has also been widely reported. Xiaojia Xu et al., in an article published in the journal Chemical Science RSC, utilized the neutral additive 4(1H)-pyridethione (4-PT) in the precursor solution of the CsPbI3 structure, achieving the black phase with a high degree of stability by delaying the crystalline structure formation process. This led to a larger grain boundary and consequently lower energy that could cause structure collapse. This effect was attributed to the ability of 4-PT to specifically coordinate with Pb^2+^ through sulfur, resulting in the formation of more ordered and compacted crystals [83].

A similar effect was reported by Yang Zhao et al. from Fudan University in Shanghai, where the additive 1,2-dimethyl-3-acetyl-imidazolium iodide (DMAII) was used. The C=O group in DMAII acted as a selective ligand to Pb^2+^, passivating the perovskite layer and suppressing electron–hole recombination. This led to the achievement and stability of the black phase, even under high-humidity and -temperature conditions, as well as an improvement in PCE from 10.8% to 13.14% compared to that of CsPbI_3_ perovskite cells without the additive present [84].

Yao Z. et al. achieved the stabilization of the black phase through doping with 2% manganese at the B-site in CsPbI_3_ perovskites. Due to the smaller radius of the Mn^2+^ ion compared to that of Pb^2+^, it occupied the interstitial spaces of the crystal lattice, facilitating the formation of the cubic phase through the tolerance factor (Goldsmidt factor). This resulted in a more stable α-CsPbI_3_, even in the presence of humidity and temperature [85].

## 4. Low-Lead and Lead-Free All-Inorganic Perovskites

The toxicity of lead is reported by different researchers as one of the obstacles for the commercialization of Pb-containing perovskite solar cells [86,87,88,89]. Despite the low amount of lead used, there is still no consensus in the scientific community regarding the real toxicity of Pb-containing PSCs. Several studies have reported the possibility of minimizing the content of Pb via partial replacement with other ions such as Sn, Ge, Bi, or Cu [90]. Other works have already focused on the complete replacement of Pb, often called “Lead-free perovskite”. However, the replacement of Pb can considerably decrease the efficiency of the photovoltaic system as it alters the optical–electronic properties. The primary point to consider when replacing Pb is to select a cation that has a similar electronic configuration. So far, the most studied element was tin (MASnI_3_), as reported by Noel et al. [91]. Since all-inorganic perovskites have presented themselves as a potential alternative to increase stability due to organic group substitution, replacing Pb and, at the same time, increasing system efficiency is a great challenge [92,93,94,95,96,97].

In addition to the similar electronic configuration, Goldschmidt’s (*t*) tolerance factor must be between 0.75 and 1 for the perovskite (ABX_3_) structure to be stable. Equation (1) shows how this factor is estimated from the size of structure ions, and Table 1 lists the ionic radius of lead and alternative cations to replace it in site B. Considering the radius of Pb in the order of 1.19 Å, it is interesting to choose ions with a similar radius (*R_B_*).
(1)$$t=\frac{\left(R_{A}+R_{B}\right)}{\sqrt{2}\;(R_{A}+R_{X})}$$
where *R_A_*, *R_B_* and *R_X_* are the ionic radius of ions *A*, *B* and *X*, respectively.

As a proposal for lead-free perovskites, Zhang Zeyu et al. suggested, in an article published in Nature Communications, the use of double perovskite Cs_2_AgBiBr_6_ with interstitial hydrogen atoms as the photoabsorbing layer in solar cells. This perovskite exhibited a high PCE value (6.37%) and good stability in the presence of humidity (85%) and high temperatures (85 °C) compared to that of lead-free perovskites reported in other studies. It maintained 60% of its initial efficiency after 1400 h under the reported conditions [98].

### 4.1. Tin-Based Perovskites

Tin is a group 14 element of the carbon family and has an ionic radius of Pb^2+^, thus exhibiting an electronic structure with properties close to those of lead, being the first metal considered an alternative to lead in perovskite halides. Furthermore, Sn-based perovskites exhibit a narrower band gap than that of their Pb analogues [99,100,101], making them promising materials in light-absorbing devices for optoelectronic applications, such that the number of publications focusing on perovskites based on in Sn has increased year after year [78,102,103].

Aifei Wang et al. [104] reported on a strategy to solve the problem inherent with bivalent lead (Pb^2+^) using tetravalent tin (Sn^4+^) to synthesize stable Cs_2_SnI_6_ perovskite nanocrystals. In his work, Wang A. obtained Cs_2_SnI_6_ nanocrystals, synthesized via a simple hot injection process using non-toxic and low-cost commercial precursors. Simulation studies carried out revealed that Cs2SnI6 nanocrystals present a well-defined face-centered cubic (FCC) structure derived from perovskite. Nanocrystals were prepared from quantum dots with different formats (spherical, nanospheres, nanowires, nanoribbons and nanoplates), as shown in Figure 8. Deformities in spherical particles were corrected via transmission and scanning electron microscopy (Cs-corrected-STEM). Field effect transistors (FET) based on Cs_2_SnI_6_ nanocrystals processed in solution exhibited high hole mobility p-type semiconductor behavior (>20 cm^2^/(Vs)) under ambient conditions. However, one of the disadvantages identified for Sn-based perovskites is their high sensitivity to moisture and oxygen, which generate oxidative processes promoting the Sn oxidation state from 2+ to 4+ (“self-doping effect”), resulting in a decrease in stability and consequently loss of efficiency over time [105]. 

Several strategies have been developed to minimize oxidation problems, which include the use of additives and atmosphere adjustment in the production and control of crystal morphology [106,107]. An example of crystal control is described by the group led by Aifei Wang [108], who reported for the first time the controlled synthesis of nanocages in CsSnBr_3_ perovskite nanocrystals, via a simple hot colloidal injection method. The nanocage structures were obtained by controlling the precursors and the reaction temperature using a self-assembly process. The synthesized CsSnBr_3_ nanocages exhibit excellent resistance to oxygen under desiccating conditions. In addition, post-treatment using perfluorooctanoic acid resulted in the better stability of CsSnBr_3_ nanocages, providing greater resistance against moisture, oxygen and light [108].

As an alternative to addressing stability loss due to oxidation in tin-based perovskites, Zheng Zhang et al. proposed the passivation of the perovskite layer through simple steps using acetylacetone (ACAC) and ethylenediamine (EDA) via sequential liquid deposition and vapor deposition. In this process, the N-H groups present in EDA coordinated with the free Sn^2+^ on the surface, stabilizing surface charges and preventing oxidation to Sn^4+^ species. Additionally, the EDA action increased the grain size and homogeneity, resulting in a photovoltaic cell that maintained 80% of its original efficiency after a 70-day period [109].

Salts like SnI_2_, SnCl_2_ and SnBr_2_ are examples of additives for increasing PCE and the stability of CsSnX_3_ perovskites. Introducing an excess of 10% SnI_2_ in preparing CsSnI_3_ improves device performance [110]. Adding SnCl_2_ to CsSnI_3_ can form a stable and protective layer of SnCl_2_.2H_2_O, resulting in increased stability [111]. The addition of SnBr_2_ to CsSnBr_3_ perovskite reduces defect density and prevents the oxidation of Sn^2+^ into Sn^4+^ [112].

Mixed lead-tin perovskites have also been explored in the literature due to their broad-spectrum light absorption and good efficiency values in photovoltaic conversion. In the work published by Zhu H. and collaborators, a power conversion efficiency (PCE) of 15.2% was achieved for solar cells based on MASn_0.25_Pb_0.75_I_3_ perovskite, prepared using a mixture of DMF and DMSO solvents. This resulted in films with a regular surface and the formation of a preferential orientation that improved crystallization and prevented defect formation [113]. Another property that has received attention in mixed Sn–Pb perovskites is the low band gap values reported for this type of material [114], which is desirable according to the Shockley–Queisser model and is a good indicator of high performance in photovoltaic applications. As reported by Yeom K. et al. in a publication in Advanced Energy Materials, the use of urea and thiourea additives in the preparation of thin films of FA_0.5_MA_0.5_Pb_0.5_Sn_0.5_I_3_ perovskites prevented the oxidation of Sn^2+^ to Sn^4+^ due to the formation of intermediate compounds between tin 2+, which acts as a soft acid, and the additives, which act as soft bases. This resulted in a band gap of 1.24 eV and a photovoltaic device that achieved a PCE of 18.5% [115].

### 4.2. Germanium-Based Perovskites

Germanium (Ge), another element of group 14, is also considered a potential candidate for replacing Pb in perovskites, in spite of its less favorable tolerance factor. Its structure has already been evaluated in MAGeX_3_ hybrid perovskites with Cl, Br and I at site X by Sun et al. [116]. The results indicated that the replacement of Pb by Ge promoted similar hole conductivity and electrons. Figure 9 shows the band structure for three perovskites described by Krishnamoorthy et al. [117], these being one inorganic and two hybrids with different organic groups. Among these, the CsGeI_3_ perovskite exhibited the highest thermal stability. On the other hand, all Ge-containing perovskites showed poor performance due to Ge^2+^ oxidation. Chen et al. [118] synthesized CsGeX_3_ perovskites (X = Cl, Br and I) via the solvothermic method. They calculated a 4.94% PCE for CsGeI_3_ perovskites, concluding that all photovoltaic parameters were improved by minimizing the crystalline defects, which reduced the recombination of charges. 

Zhang X. et al. developed an all-inorganic CsGeI_3_ perovskite solar cell without the need for a hole transport layer (HTL). By using ZnOS as the electron transport layer (ETL) and optimizing critical parameters, they achieved a high-power conversion efficiency (PCE) of 26.70%. This work improves the performance of germanium-based perovskite solar cells while reducing costs, offering a new avenue for low-cost and efficient solar cells [119].

Sheetal Solanki et al. conducted research on perovskite solar cells, focusing on improving their efficiency and stability. They investigated the development of lead-free alternatives, particularly the cesium germanium triiodide (CsGeI_3_) heterostructure perovskite solar cell. Through simulations using the solar cell capacitance simulator (SCAPS), they achieved a conversion efficiency of 12.2% and significantly improved device parameters. The research aimed to optimize lead-free components by studying the influence of defect density and doping concentration on the absorber layer [120].

### 4.3. Bismuth-Based Perovskites

Bismuth (Bi) is a sixth period element that belongs to the nitrogen (VA) family and has an isoelectronic valence shell in relation to Pb, making it another possible substitute for lead. Furthermore, its ionic radius (1.03 Å) is similar to that of Pb (1.19 Å) [121]. Thus, Bi has been widely explored in halogenated perovskite structures with optical applications. In the A_3_Bi_2_X_9_ structure, A and X atoms are more compacted and Bi atoms occupy two thirds of the octahedral X_6_ space. In work published by Park et al. Bi-based double perovskites with an A_3_Bi_2_I_9_ configuration (A = MA^+^, Cs^+^) were prepared, in which the (Bi_2_)_3_ structures were surrounded by cations (Figure 10) [122]. In the published results, it was verified that the perovskite Cs_3_Bi_2_I_9_ presented a photoluminescence intensity five times greater when compared to that of the other two perovskites (MA_3_Bi_2_Cl_x_ and MA_3_Bi_2_I_9_). Furthermore, photovoltaic cells prepared using Cs_3_Bi_2_I_9_ as photoactive layer showed superior performance with Jsc = 2.15 mA.cm^−2^, Voc = 0.85 V, FF = 0.60 and PCE = 1.09%. It was also reported that the light absorption and PCE spectra showed little change after one month of storage (in dry air, with humidity below 10%, and in the dark) indicating good stability for this class of materials.

Korukunda T. B. et al. [116] studied the development of carbon-based perovskite solar cells (C-PSCs) and the challenges associated with perovskite infiltrating into the C-PSC stack. The authors propose a modified electric field (EF)-assisted spray technique, along with a greener solvent system, for the infiltration of bismuth perovskite (Bi-perovskite) in the C-PSC stack. The developed protocol allows for the controlled and faster infiltration of perovskite, addressing the issue in C-PSCs. The spray devices exhibit an improved efficiency of approximately 80% compared to that of drop-casted devices, and the selective plane growth of the Bi-perovskite enhances stability. Additionally, the use of a greener solvent system contributes to the process’ advantages. This work represents the first-ever application of the spray-processed non-lead perovskite with a greener solvent system in a large-area hole transport layer (HTL)-free C-PSC device [123].

The study conducted by Karthick et al. explores the introduction of Cu^+^ and Bi^3+^ into FA_0.85_Cs_0.15_Pb(I_0.85_Br_0.15_)_3_ double-cation planar perovskite solar cells. The addition of CuI and BiI_3_ in different doping ratios resulted in significant changes in the morphology and crystalline quality of the perovskite layer. While Cu doping led to increased grain dimensions, bismuth incorporation caused drastic alterations even at low doping levels. However, the efficiency of the Cu- and Bi-doped devices was found to be poorer compared to that of the undoped reference device. An analysis of charge recombination indicated faster kinetics for the doped devices, suggesting the importance of controlling defects when substituting lead with metallic cations. Preliminary aging tests also highlighted the impact of defects on device lifetime. The study emphasizes that using a monovalent cation as a substitute for lead may be less detrimental to the perovskite structure compared to the use of trivalent alternatives like Bi, especially at low doping ratios that can still achieve efficient and stable devices [124].

Bi-based perovskites have been shown to exhibit a high quantum photoluminescence yield and better stability under humid environments compared to those of Pb-based halide perovskite. Yang et al. synthesized Cs_3_Bi_2_X_9_ (X = Cl, Br, I) nanocrystals via a simple reaction at room temperature [125]. The emission wavelength was adjusted from 400 nm to 560 nm with different halide compositions. The quantum photoluminescence yield of Cs_3_Bi_2_Br_3_ was improved from 0.2% to 4.5% via the addition of an extra surfactant (oleic acid), as a result of the passivation process. In addition, the Cs_3_Bi_2_Br_9_ nanocrystals exhibited high stability in air for more than 30 days. Leng et al. studied a series of Cs_3_Bi_2_X_9_ perovskites (X = Cl, Br and I) and found that the Cs_3_Bi_2_Br_9_ composition presented excellent photostability and moisture stability due to its all-inorganic nature and the surface passivation phenomenon due to BiOBr formation [126]. Johansson et al. demonstrated the synthesis of Cs_3_Bi_2_I_9_ and suggested that film formation with the CsBi_3_I_10_ layered structure enabled the photoconversion spectrum to be extended to above 700 nm, in turn enabling increased light absorption [127].

Lou et al. prepared quantum dots of Cs_3_Bi_2_X_9_ (X = Cl, Br, I) via the stable binding of octylammonium halide (OA-X) and oleic acid [128]. Cs_3_Bi_2_Br_9_ exhibited quantum photoluminescence yields of up to 22% with an emission peak at 414 nm and a full width at half maximum of 38 nm, and could remain stable at up to 180 °C. In addition, Cs_3_Bi_2_Cl_9_ and Cs_3_Bi_2_I_9_ could be easily obtained via a fast-reversible anion exchange reaction, exhibiting fluorescence emission wavelengths ranging from 380 nm to 526 nm with a quantum photoluminescence yield of 62% and 2.3%, respectively.

### 4.4. Antimony-Based Perovskites

Another possible candidate for lead substitution is the trivalent antimony (Sb), which has a pair of isolated 5s^2^ electrons [129,130]. Furthermore, Sb is more than twice as cheap as Sn [131]. The A_3_Sb_2_X_9_ structure is the same as that of A_3_Bi_2_X_9_, where Cs_3_Sb_2_I_9_ also has the 0D dimer form and the layered 2D form. The absorption peak can be adjusted from 558 nm to 453 nm with an increase in bromine content [132].

Briefly, 2D layered Cs_3_Sb_2_I_9_ thin-films were prepared via a two-stage deposition approach as reported by Saparov et al. (see Figure 11), exhibiting an indirect 2.05 eV band gap and improved stability against environmental conditions compared to that of MAPbI_3_ [133]. However, Cs_3_Sb_2_I_9_-based solar cells exhibited a V_oc_ value of 0.25 to 0.30 eV, which is lower compared to that of their hybrid analog (CH_3_NH_3_)_3_Sb_2_I_9_ (0.89 V) [134].

As for stability, in a recent study published by Chonamada et al. [135], the degradation of Cs_3_Sb_2_I_9_ polymorphs (dimer and layer forms) was evaluated under conditions of water, light, and elevated temperature, which are the main factors reported as causing degradation in halogenated perovskites. Using X-ray diffraction and thermogravimetry techniques, it was confirmed that the layered polymorph structure presented greater stability when compared to that of the dimer polymorph. In the work, it was reported that the complete degradation of the dimeric structure occurred in 49 days, whereas in the stratified form, the complete degradation occurred in 88 days, although both Cs_3_Sb_2_I_9_ polymorphs are relatively more stable in relation to the tested factors than are the organic–inorganic halide perovskites. However, it was found that the main cause of degradation in Cs_3_Sb_2_I_9_ occurs due to the diffusion of iodine in the structure, and this process is accelerated due to the high reactivity of antimony iodide (SbI_3_) in oxygen. Light, water and heat conditions also cause the degradation of Cs_3_Sb_2_I_9_ and, therefore, the use of this material for applications in an ambient atmosphere would require adequate encapsulation or other necessary measures.

The performance of the Cs_3_Sb_2_I_9_ polymorph acting as a photoactive layer in photovoltaic devices was evaluated and discussed in a work carried out by Singh A. et al. As a result, it was verified that the binding energy and lifetime of the electron–hole pair generated from the perovskite Cs_3_Sb_2_I_9_ in the layered form were approximately 100 meV and 6 ns, respectively, obtaining better results in relation to the photovoltaic properties when compared to those of the polymorph dimeric. The solar cell prepared using Cs_3_Sb_2_I_9_ as the photoabsorber layer exhibited an open-circuit voltage of 0.72 V and an energy conversion efficiency of 1.5%, which is the highest reported for a fully inorganic Sb-based perovskite so far [136].

Tang G. et al. [137] presented a strategy to prepare layered halide Cs_3+n_M(II)_n_Sb_2_X_9+3n_ double-perovskites, partially doping the A site with Sn or Ge. Relevant photovoltaic properties were achieved by inserting octahedral [MX_6_] layers (M = Sn or Ge) based on principles of increased electronic dimensionality. Compared to Cs_3_Sb_2_I_9_, the addition of octahedral layers of [SnI_6_] or [GeI_6_] into the [Sb_2_I_9_] bilayers allowed the obtention of more suitable band intervals, smaller carrier effective masses, larger dielectric constants, lower exciton binding energies and higher optical absorption. In practice, structural modification can result in band gap values of 1.66 and 1.76 eV after doping with Sn (II) and Ge (II), respectively. 

A site alteration in A_3_Sb_2_I_9_ perovskites containing antimony was also investigated in a study published by Correa J. et al. [138], who reported the effects of changing the photovoltaic properties of antimony-based compounds by replacing Cs with Rb and K. The band gap value, structure, carrier lifetime, film morphology and performance of the photovoltaic device were determined experimentally. In addition, computer simulation was performed using density functional theory (DFT) to calculate the lowest energy structural conformation, band structures, effective carrier masses and phase diagrams. Thus, the structural and optoelectronic properties of the material change as a function of the cation that makes up the A site of the structure were analyzed. It was reported that the perovskite Cs_3_Sb_2_I_9_ has a 0D structure and a higher electron–hole pair binding energy (175 ± 9 meV), in addition to having an indirect band gap and, when applied to a solar cell as a photoactive layer, it presented a low photocurrent (0.13 mA cm^−2^). The Rb_3_Sb_2_I_9_ structure, on the other hand, has a 2D structure, direct band gap and a lowest exciton binding energy (101 ± 6 meV) among the investigated materials, having the highest photocurrent (1.67 mA cm^−2^) when applied to a photovoltaic device. The material prepared with K composing the A site (K3Sb2I9) maintained the structural configuration, but with intermediate exciton binding energies (129 ± 9 meV) and intermediate photocurrents (0.41 mA cm^−2^). Despite remarkably long lifetimes in all compounds (54, 9, and 30 ns for Cs, Rb, and K-based materials, respectively), low photocurrent values limited the performance of all devices. In conclusion, the authors attested that carrier recombination is limited by large exciton binding energies (observed experimentally) and large effective carrier masses (calculated from the density functional theory). Thus, the highest efficiency recorded (0.76%) was observed in the compound containing Rb in the A site, which has a direct band gap, lower exciton binding energy and a lower calculated effective mass of the electron. In conclusion, the authors recommend that, in order to obtain more accurate results from the opticoelectronic properties in photoactive layers of lead-free devices, such as exciton binding energies and effective masses, faster and more robust computational tools are needed [138].

In recent years, the scientific community has shown great interest in the numerical simulation of lead-free perovskite solar cells. All-inorganic cesium antimony iodide (Cs_3_Sb_2_I_9_) has been explored as a light-absorbing layer in the development of lead-free perovskite solar cells. Ahmad K. et al. [139] reported the numerical simulation of lead-free perovskite solar cells using Cs_3_Sb_2_I_9_ as the light absorber layer, achieving an excellent efficiency of 12.54% using SCAPS-1D software. The optimized lead-free perovskite solar cells were fabricated using the FTO/TiO_2_/Cs_3_Sb_2_I_9_/spiro-OMeTAD/Au device architecture. The authors also synthesized Cs3Sb2I9 films and examined their physiochemical and optical properties. The fabricated lead-free perovskite solar cells demonstrated a decent efficiency of 1.07% with an excellent Voc of 0.622 V. Although the efficiency was relatively low compared to that of the optimized simulated lead-free perovskite solar cells, the authors suggest that it can be further enhanced by introducing novel charge extraction or transport layers [139].

Liang J. et al. [140] report the successful fabrication of totally inorganic solar cells containing lead-free perovskite (PSC) that showed good stability, had low costs and contained materials that cause less of an impact on the environment. Cs_3_Sb_2_I_9_ perovskite films that showed a well-established morphology, long lifetime, low defect density and defined band structure with other functional layers were prepared via a solution-based method. Alternatively, carbon electrodes were used in the work instead of the labile organic components and expensive noble metal electrodes commonly seen in traditional PSCs, resulting in the champion PCE of 1.43%, which is more than twice as high as the results of that from previous reports. Low-cost, Pb-free, all-inorganic PSCs have demonstrated excellent stability under various extreme conditions, making them promising candidates for next-generation commercial PSCs [140].

### 4.5. Titanium-Based Perovskites

Titanium (Ti) is one of the most abundant elements on Earth, non-toxic and ultra-stable, and has been reported as a promising candidate for Pb replacement [141], despite Ti^4+^ being a much smaller cation. The structure A_2_TiX_6_ is a double-perovskite structure with a vacancy similar to that of A_2_SnX_6_. Ju M. et al. [141] prepared Cs_2_TiI_x_Br_6-x_ perovskite, and its band gap could be adjusted from 1.02 eV to 1.78 eV. In addition, studies were carried out to evaluate the substitution of the A-site element on the electronic properties of the Cs_2_TiI_6_, Rb_2_TiI_6_, K_2_TiI_6_ and In_2_TiI_6_ perovskites. It was found that the size of the radius of the cation that makes up the A site in the perovskite structure directly influences the band gap range of the material, indicating that the angle of inclination of the octahedron [BX_6_]^2−^ affects the electronic structure of the material [141]. In another work, the same group also synthesized Cs_2_TiBr_6_ perovskites via a two-step deposition method [142]. Thin Cs_2_TiBr_6_ films showed a favorable band gap of 1.82 eV, suitable energy levels, and superior intrinsic and environmental stability. The contact with C_60_ (buckyball) facilitated electron transfer from Cs_2_TiBr_6_ to TiO_2_. Therefore, Cs_2_TiBr_6_-based solar cells achieved a 3.28% PCE with V_oc_ = 1.02 V, FF = 0.56, and J_sc_ = 5.69 mA cm^−2^. In addition, thin Cs_2_TiBr_6_ films exhibited greater tolerance to heat, humidity, and light when compared to MAPbI_2_Br. 

### 4.6. Copper-Based Perovskites

Copper (Cu) is a metal that belongs to the transition group and has an electronic configuration, 3d^9^ (t^6^_2g_e^3^g). This element has drawn considerable attention due to the high stability of the oxidized state, in addition to the high absorption coefficient in its compounds. Yang et al. reported the quantum dot synthesis of totally inorganic perovskites using Cu at the B site and the combination of different halogens at the X site (Cs_2_CuX_4_, X = Cl, Br, and Br/I) [143]. In the described material, the anion exchange reactions allowed the adjustment of photoluminescence peaks from 385 nm to 468 nm., since the band gap of the material is directly related to the size of the particles, in which the alteration of the precursor proportions provides an adjustable band gap. These copper-based perovskites exhibited excellent stability in the presence of oxygen compared to other lead-free perovskites, maintaining 92% of the original photoluminescence intensity after storage under ambient conditions for 30 days. Even after 5000 min of ultraviolet light irradiation, 34.2% of the initial photoluminescence intensity was maintained.

As there are few studies in the literature focusing on Cu-based inorganic perovskites, studying Cu-based hybrid perovskites can provide relevant information that contributes to the design of inorganic perovskites. In the series of (CH_3_NH_3_)_2_CuCl_x_Br_4−x_ perovskites, the role of Cl was found to be essential for stabilization, while the Br/Cl ratio affects optical absorption, which can be extended to the near-infrared region for optimal spectral overlap with solar irradiance. However, even in hybrid perovskites, performance is compromised by the low absorption coefficient of these materials and the large mass of holes.

### 4.7. Double Perovskites and Bulk Heterojunction

Recent work found that it is possible to replace the Pb^2+^ cation by combining a monovalent M^+^ cation and an M ^3+^ cation to form AMM’X_3_ 3D double perovskites (A = Cs, CH_2_NH_3_; M = Ag; M’ = Bi, Sb; X = Cl, Br, I) [144,145,146]. The perovskite duo Cs_2_AgBiBr_6_ with an indirect band gap of 1.95 eV was reported by Slavney et al. [147]. It has been found that the lifetime of photoluminescence is shorter in powders when contrasted with that of single crystals, due to the increased presence of defects and surface states in powders. This suggests that short-lived and intermediate processes may arise from capture and/or emissions from surface states, respectively. In addition to showing better stability in hot and humid conditions, the Cs_2_AgBiBr_6_ perovskite also obtained a longer charge carrier recombination time when compared with that of MAPbI_3_ films [147].

Some studies focus on the combination of different double perovskites to form heterojunctions, which consist of an interface between two solid state materials. In a broader sense, it includes crystalline and amorphous structures made of metallic and insulating materials, fast ion conductors, and semiconductors. The first article to make use of heterojunctions in photovoltaic cells was published in 1983 by Okuda K. et al. [148]. The authors developed an amorphous silicon solar cell stacked with polycrystalline silicon. A conversion efficiency of above 12% was obtained with a cell structure of ITO//n-i-p a-Si//n a-Si/p poly c-Si//Al [148]. The use of heterojunctions has received attention from the scientific community and has been applied in many fields, not only in solar cells but also in the manufacture of lasers, bipolar transistors, and field effect transistors, as illustrated in Figure 12 [148,149]. Heterojunction solar cells have been widely reported for organic, inorganic, and hybrid photovoltaic devices, but a heterojunction solar cell containing double perovskite without the presence of lead has only recently been reported. High-quality Cs_2_AgBiBr_6_ double-perovskite film was manufactured through a low-pressure solution processing method under ambient conditions and showed an energy conversion efficiency of 1.44% [149]. Due to the high stability of the photovoltaic device when exposed to ambient conditions of heat and humidity, even without encapsulation, combined with the ease of producing a high-quality film, this material has shown promise as an option for lead-free third-generation photovoltaic technology [149]. Recently, Hu W. et al. [150] developed a bulk heterojunction bismuth-based perovskite solar cell with photoactive layers of Cs_3_Bi_2_I_9_ and Ag_3_Bi_2_I_9_ perovskites, achieving a record efficiency of approximately 3.6%. The beneficial effect was accomplished from the increased crystal grain size of Cs_3_Bi_2_I_9_ and optimized grain orientation of Ag_3_Bi_2_I_9_. The devices were tested for 450 h at a temperature of 85 °C in a controlled atmosphere system (glove box). Surprisingly, the device maintained approximately 90% of the initial performance during the test, indicating excellent stability [150]. The development of heterojunctions, therefore, will be a trend in future works, since there is a search for alternative materials and new methods that seek heterojunctions with an abrupt and clean interface according to the desired band structure, aiming at the formation of quantum dots suitable for the assembled system. 

The review article published by Fan Q. et al. [151] highlights the need to replace lead in the preparation of halide perovskite nanocrystals, presenting strategies for enhancing optical properties and stabilities of perovskite under ultra-violet radiation, heat, and humidity. Monovalent and trivalent elements such as Ag^+^ and Bi^3+^ are cited as examples woth with which to replace Pb and form Cs_2_AgBiX_6_ double perovskites. Finally, the authors cite three strategies for developing more stable perovskites: (1) coating the perovskites with organic or inorganic components; (2) developing “defect passivation” methods and exploring degradation mechanisms; (3) replacing Pb with other elements more resistant to air, for example Fe or Zn [144]. In other work, Dave K. et al. [87] reported a review focused on the double perovskites A_2_B_(I)_B_(III)_X_6_, A_3_B_(III)2_X_9_ and A_2_B_(IV)_X_6_, highlighting the structures, doping, and applications in solar cells, LEDs, photodetectors, and photocatalysts. The emphasized perspectives, according to the authors, are to increase the photoluminescence yield and to passivate surface defects in order to increase the stability of the perovskite structures [87]. In a recent paper, Parida et al. [102] review the recent progress in fabricating CsPbX_3_ and CsSnX_3_ inorganic perovskite solar cell devices with high PCE and stability, highlighting various strategies that include modifying the composition, solvent engineering, deposition techniques, and surface and interfacial passivation.

Among the materials studied, bismuth- and silver-based perovskites can be highlighted [150,151,152]. The perovskite Cs_2_AgBi_0.75_Sb_0.25_Br_6_ has been studied and applied in the development of tandem-type solar cells in combination with perovskite FACsPb_0.5_Sn_0.5_I_3_. In general, the study conducted by Madan et al. [152] revealed that the tandem-type solar cells showed a conversion efficiency of 17.3% due to an improvement in the open-circuit voltage (VOC), which was 1.83 V. Hu W. et al. [150] reported a considerable increase in energy conversion efficiency through the combination of two perovskites: Cs_3_Bi_2_I_9_ e Ag_3_Bi_2_I_9_. The heterojunction formed resulted in an efficiency of 3.1% due to the grain control present in the nanocrystals.

The perovskite Cs_2_AgBiBr_6_ with Br at the X site was recently studied via two different approaches [153,154]. In the first one, the charge transfer to TiO_2_ and ZnO oxide semiconductors was investigated. The results indicate that the accumulation of holes induces the oxidation of quantum-dots, resulting in the photodegradation of the perovskite film. Thus, the authors indicate that the development of an appropriate HTL can improve the stability of perovskites [153]. The in situ high-pressure photoluminescence (LP) technique was also used to evaluate the band variation as a function of structural change under extreme conditions. The results obtained showed an emission varying from 520 to 1000 nm after the crystalline system changed from cubic to tetragonal due to the distortion of the AgBr_6_ and BiBr_6_ octahedrons during compression when a pressure of 4.8 GPa was exerted. In addition, the obtained band gap value did not change after pressure release under ambient conditions. The results provide a feasible strategy with which to extend the spectral coverage of lead-free double-perovskite materials [154]. 

Studies incorporating Cl at the X site in Cs_2_AgBiBr_6_ perovskites have been conducted by Kumar Chini et al. [155]. The authors have performed computational studies and synthesized and characterized mixed halide-based double perovskites in order to investigate the incorporation of Cl in Cs_2_AgBiCl_x_Br_6−x_ (x = 0 to 6). The results indicate that the band gap is strongly dependent on the nature of the halide ions constituting the BiX_6_ octahedra. When the BiX_6_ octahedra contain both chlorine and bromine ions, the band gap increases linearly with the increase in the chloride content. However, when the octahedra contain only one halide ion (Br or Cl), the band gap remains nearly equal to that of Cs_2_AgBiBr_6_ with up to 50% Cl doping. According to the authors, these materials are promising photovoltaic absorbers owing to their favorable band gap (~1.98 eV) and high stability in ambient conditions [155]. Other studies have incorporated indium combined with chlorine to form the double perovskite Cs_2_AgInCl_6_. Despite the high band gap (3.01 eV), the authors, based on computer simulations, considered this perovskite a potential for photovoltaic applications, since the absorption spectrum is in the visible region, in addition to the good results regarding reflectivity and refraction [156].

Antimony, tin and germanium-based perovskites have recently been investigated in studies that demonstrate their thermoelectric properties and degradation phenomena [135,157]. From structural simulation studies, the substitution of lead with tin and germanium was evaluated, and the authors identified that perovskites with a high content of Cs(Ge,Sn)I_3_ have better thermoelectric properties [157]. On the other hand, experimental degradation studies were carried out on the Cs_3_Sb_2_I_9_ perovskite using X-ray diffraction and thermogravimetric analyses, revealing that the degradation phenomenon is associated with the formation of dimers in the presence of humidity, light, and high temperatures. The reactivity of SbI_3_ with oxygen accelerates the process of degradation of perovskite [135].

Titanium-based perovskite has also received attention in the literature, with a structure of A_2_TiX_6_. Kong D. et al. [158] synthesized thermally stable cesium titanium halide perovskite (Cs_2_TiX_6_, X = Cl, Br) crystals and films, through a full aqueous solution process at room temperature. Structural, thermal, and optical properties were investigated experimentally and the results obtained were compared with those of computationally generated theoretical models using first-principle calculations. The produced materials demonstrated a tunable, quasi-direct band gap ranging from ~1.7 eV (for Cs_2_TiBr_6_) to ~2.5 eV (for Cs_2_TiCl_6_), and photoluminescence peaked from ~535 nm (Cs_2_TiCl_6_) to ~670 nm (for Cs_2_TiBr_6_). Furthermore, the materials are stable in an ambient environment up to ~500 °C [158]. The partial replacement with Br of Cl (Cs_2_Ti(Br_1−x_Cl_x_)_6_ was evaluated through computer simulations by Li et al. to explore the electronic and optical properties [152]. According to the results, the formation energies are reduced with an increase in the Cl content, while the stability of the systems increases and further extends the lifetime. The perovskite Cs_2_Ti(Br_0.75_Cl_0.25_)_6_ presented a favorable band gap. Compared to Cs_2_TiBr_6_, chlorine-doped perovskites possess more suitable band intervals and lower formation energies, while iodine-doped systems result in better optical performance, but the systems become unstable. The perovskite Cs_2_Ti(Br_0.75_Cl_0.25_)_6_ is considered an ideal candidate for perovskite solar cells [159].

Finally, despite the few studies reported in the literature, oxide-based perovskites have gained prominence in the literature. Li et al. [160] studied oxide-based perovskite (K, Bi) (Nb, Yb)O_3_ with ferroelectric and photovoltaic properties, leading them to suggest a potential application. Despite the band gap of approximately 1.45 eV, the energy conversion efficiency was 0.85%, measured using a solar simulator (AM 1.5G), indicating that performance can be improved by controlling charge recombination.

## 5. Degradation Mechanisms

A significant effort has been undertaken by various authors with the goal of assessing the different degradation mechanisms of perovskites acting as the photoabsorbing layer in photovoltaic cells, aiming to optimize the performance and lifespan of these materials. In an article published in Nature Communications, Ahn N. and collaborators [161] evaluated the effects of humidity on the degradation of cells containing hybrid perovskites and associated these effects with charges trapped in structural defects, leading to the irreversible formation of species that interact and degrade organic cations. In the same work, the authors describe that such mechanisms can be divided into two stages. The first stage involves water absorption, forming hydrated perovskites such as CH_3_NH_3_PbI_3_·H_2_O or (CH_3_NH_3_)_4_PbI_6_·2H_2_O, which exhibit a lower degree of interaction between organic cations and PbI_6_^4−^ octahedra. The second stage involves the deprotonation of organic cations due to the formation of an electric field induced by charges trapped in defect sites, generating species that volatilize into the environment, as in the case of MA (CH_3_NH_3_^+^ → CH_3_NH_2_ ↑ + H_3_O^+^). In the described process, the key point is the combination of light presence, responsible for trapping charges, and the presence of water. Degradation tests were conducted in a high-humidity environment (RH = 90%) without light, revealing that effects caused solely by water were reversible. However, the combination of radiation with 20% humidity resulted in irreversible effects on the photovoltaic cell [161]. Mahiny M. and collaborators, through computational simulations and experimental tests, obtained similar results regarding the degradation mechanism induced by trapped charges. They found that grain boundary regions act as sources of reactive species for oxygen and ambient water, creating non-radiative recombination centers where degradation processes are initiated [162].

In addition to humidity, oxygen also has a significant influence on the degradation of perovskite-based solar cells, as reported by Aristidou N. et al. [163] They observed that O_2_ is adsorbed on the perovskite surface and diffuses through halogen vacancy sites (I^−^, Br^−^, and Cl^−^), generating energy states between the conduction and valence bands that trap charges when the material is photoexcited. This leads to the formation of highly reactive superoxide species (O_2_^−^) causing the production of CH_3_NH_2_, PbI_2_, I_2_, and H_2_O [163]. Temperature also plays a crucial role in the perovskite degradation process. While it alone does not cause the loss of stability in the photoabsorbing layer, it works in conjunction with reactions promoted by oxygen and ambient humidity, acting as a facilitator due to increased reaction kinetics, resulting in accelerated degradation processes [164,165,166]. 

The degradation effects were also evaluated in fully inorganic perovskites, as demonstrated in a study by Zhu W. and collaborators, investigating the degradation process of Cs_2_SnI_6_ perovskite. The material exhibited stability when exposed to environments with up to 80% RH; beyond this, the formed crystals started undergoing dissolution processes, generating the respective iodide salts (CsI and SnI_4_). Surface defects in the crystal acted as energetically favorable sites in the process [167]. Similarly, Yao W. et al. obtained comparable results for CsPbI_2_Br perovskites, where for RH values above 50%,e α-CsPbI_2_Br (the black phase) decomposed into δ-CsPbI_2_Br (the yellow phase) and later into PbI_2_. These processes are initiated at the grain boundaries of thin films, with generated ions migrating into the grain interiors due to a potential difference [168]. 

## 6. Encapsulation

The low relative stability of the device in the long term still significantly hinders the industrial commercialization of this photovoltaic technology. For this reason, it is imperative to address stability issues and simultaneously achieve high efficiency and durability. One option to enhance the durability of perovskite-based photovoltaic devices is encapsulation. Despite intrinsic instability, encapsulation is considered an effective method to improve the long-term stability of these devices under environmental conditions. Therefore, detailed requirements for the encapsulation process and encapsulating materials are necessary to meet the needs of environmentally sensitive perovskite solar cells (PSCs), especially with regard to temperature, oxygen, humidity, and UV light.

In a recent work published by Xiang Ling and collaborators, the latest progress in encapsulation techniques is presented. Specifically, three encapsulation strategies are discussed. The glass-to-glass encapsulation strategy originates from conventional silicon photovoltaic technology, where the solar device is sandwiched between two glass sheets using an encapsulating adhesive. This strategy allows the retention of 95% of the energy conversion efficiency of silicon solar cells after 20 years and is currently one of the most widely used encapsulation strategies to improve the long-term stability of perovskite devices. Suitable materials must be selected as encapsulating adhesives to achieve excellent encapsulation effects, as it is crucial to minimize damage from the encapsulation process and materials to the perovskite in this strategy. As mentioned above, cover glass can meet the requirements of high transparency, good mechanical properties, and excellent moisture stability, although it is not suitable for flexible devices. For such cases, the polymer encapsulation technique is more suitable. Currently, thin-film encapsulation techniques are being studied using various application methods already described in the literature [169].

Raman K. R. and collaborators, in their work, describe, among other properties, the requirements for encapsulation. The reader of this work, venturing into the realm of PSCs, is invited to explore this author’s review. In addition to presenting the factors affecting cell stability, the authors delve into optical properties, mechanical, and chemical requirements, as well as characteristics of the oxygen transmission rate and water vapor transmission rate that must be assessed to achieve satisfactory results in the encapsulation of devices [170].

Aitola K. and collaborators have published work in conducting an exploratory and well-discussed survey on encapsulation materials and methods. They cover established techniques used in the crystalline silicon photovoltaic industry, such as layers of ethylene–vinyl acetate combined with front and back glass sheets and a polyisobutylene edge sealant, to experimental approaches employed in academic research on emerging solar cells. These include fluoropolymeric composites that have yielded promising stability results, and additional functionalities that can be combined with encapsulants [171]. In brief, encapsulation materials may involve encapsulants with additional functionalities and bio-based raw materials

## 7. Final Considerations

The development of perovskite-based photovoltaic devices has indeed significantly advanced in the last 10 years since the first lead-based hybrid perovskites. Great efforts have been made to increase efficiency (PCE) and resistance to temperature and humidity degradation based on perovskite structural modification and architectural change. Due to the unresolved problems, particularly concerning device stability, all-inorganic perovskites have gained prominence, although with fewer reports in the literature. CsPbI_3_ perovskite has potential among inorganic perovskites, but suffers from degradation problems due to black phase instability (α-CsPbI_3_), constituting problems that still need to be solved. Motivated by the potential problems associated with toxicity, Pb has been totally or partially replaced by elements such as Ge, Sn, Sb, Bi, Cu, or Ti. In addition, maintaining the structural quality of films in large-scale production is considered a major challenge. The development of all-inorganic perovskites was preceded by significant advances from hybrid perovskites with high levels of efficiency when Pb is present in the structure. However, this efficiency is at most 10% in inorganic perovskites, while in hybrid perovskites it exceeds 25% efficiency. Because lead is a toxic element, its application is limited. In the last five years, many efforts have been required to replace lead-based materials, which include perovskites based on Sb, Bi, Ge, Ti, and Sn. However, stability problems have been reported in some systems, such as Sb- and Sn-based perovskites, due to metal oxidation and the formation of dimers. Although still not very efficient, some studies also pointed out the use of oxide perovskites as a photoactive layer. Naturally, much research has yet to be conducted for inorganic perovskites to increase their potential applications. Some strategies to solve the main problems include new methods of synthesis, the manipulation of structure and morphology, and the configuration of the architecture cell, in order to improve the efficiency and stability of the devices.

## Figures and Tables

**Figure 2 nanomaterials-14-00332-f002:**
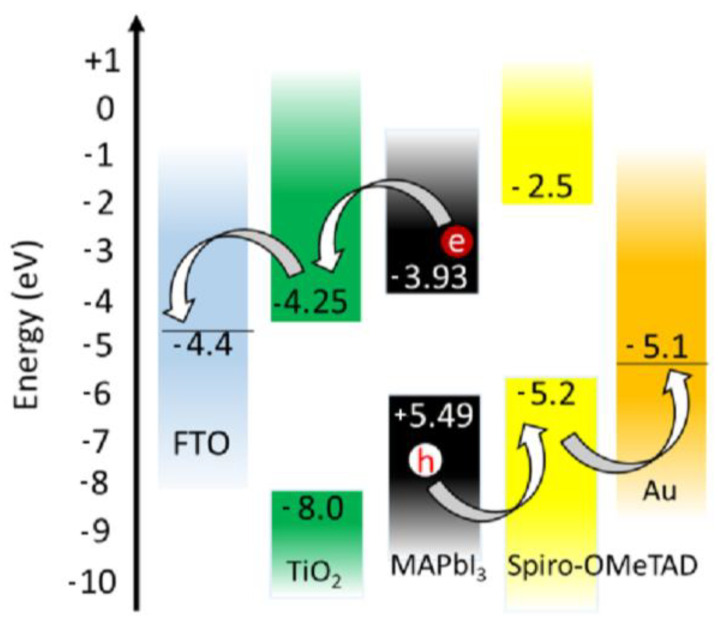
Typical energy diagram of a hybrid perovskite PVC using an MAPbI_3_ structure as a light absorber, TiO_2_ as an ETL and Spiro-OMeTAD as an HTL. FTO and Au are top and bottom contacts. Adapted from [43].

**Figure 3 nanomaterials-14-00332-f003:**
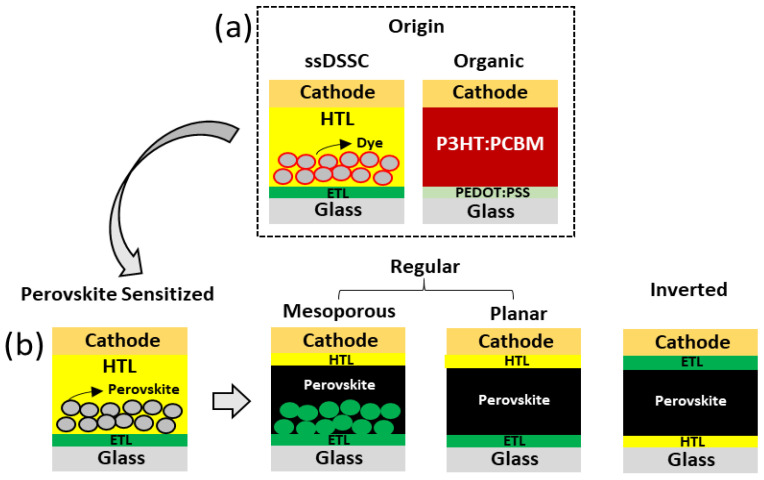
(**a**) Solid-state-dye-sensitized solar cell (ssDSSC) and regular organic solar cell. (**b**) Different configurations of perovskite solar cells (from left to right): sensitized perovskite solar cell; mesoporous perovskite solar cell; planar perovskite solar cell; and inverted perovskite solar cell. Adapted from [44,45].

**Figure 4 nanomaterials-14-00332-f004:**
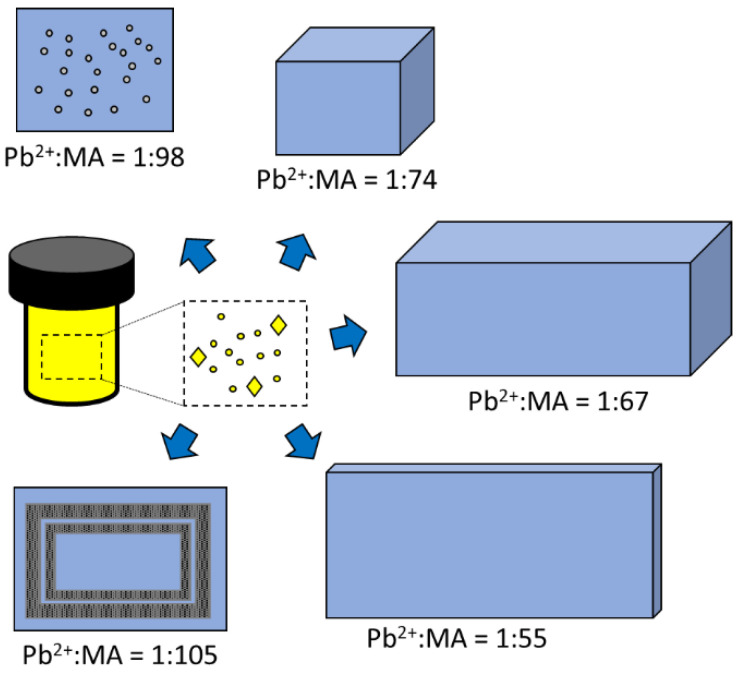
Large CH_3_NH_3_PbBr_3_ single crystal sheets synthesized via the one pot solvothermal method. The crystalline surface was recorded via the controlled local accumulation of methylammonium cations. Adapted from [67].

**Figure 5 nanomaterials-14-00332-f005:**
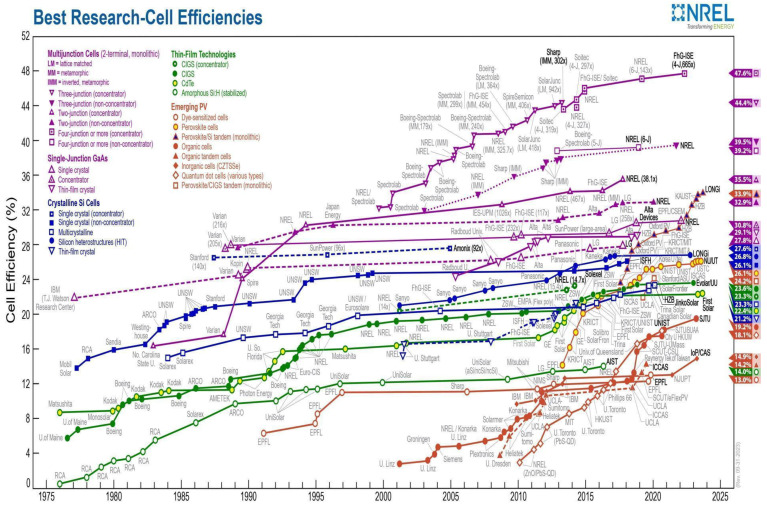
Evolution of the efficiency of solar cells based on perovskites provided by National Renewable Energy Laboratory (NREL) [28].

**Figure 6 nanomaterials-14-00332-f006:**
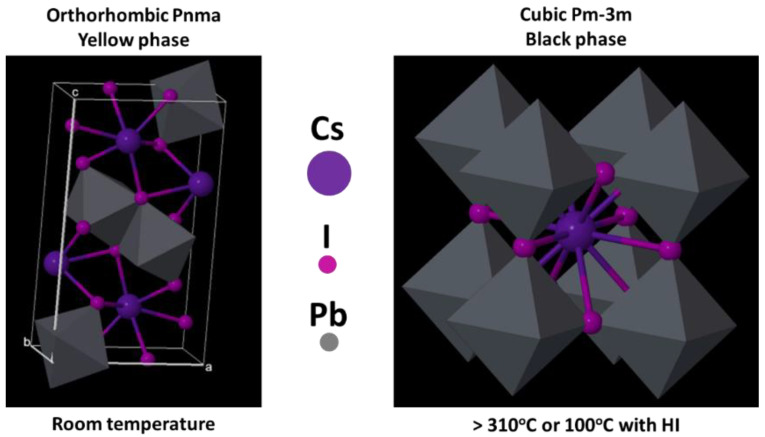
Illustration of crystalline structure change (yellow → black) in CsPbI_3_ perovskite. Source: [77].

**Figure 7 nanomaterials-14-00332-f007:**
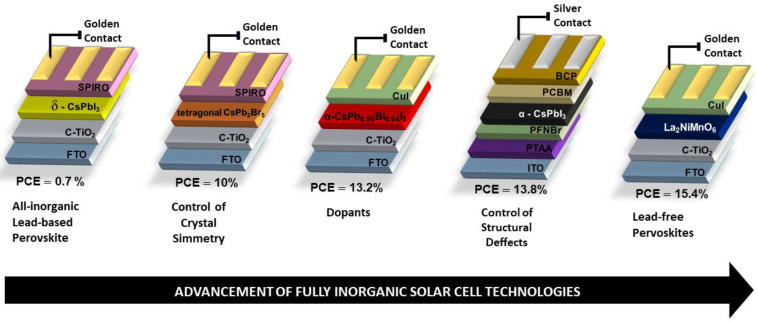
Representation of fully inorganic perovskite-based solar cells demonstrating some of the methodologies employed by various authors for improvements in the performance of the photovoltaic device [70,81,82,83].

**Figure 8 nanomaterials-14-00332-f008:**
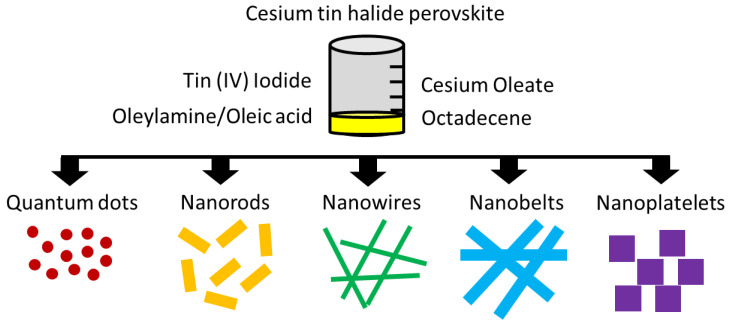
Cs_2_SnX_6_ (X = Cl, Br, I) with different morphologies. Adapted from [104].

**Figure 9 nanomaterials-14-00332-f009:**
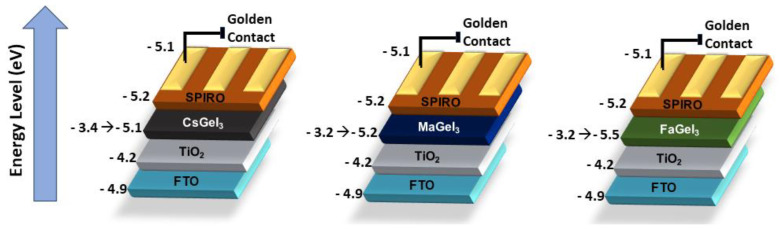
Energy level diagram for solar cells containing different Ge-based perovskites. Adapted from [117].

**Figure 10 nanomaterials-14-00332-f010:**
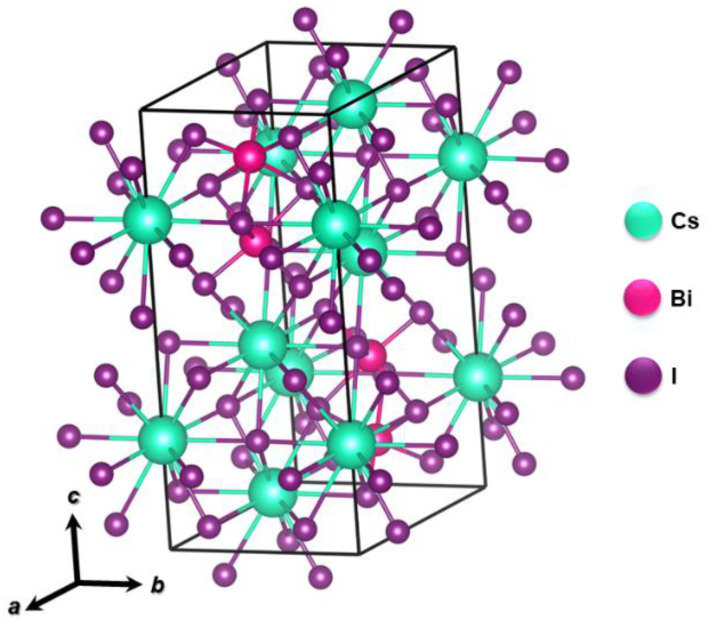
Crystalline structure of Cs_3_Bi_2_I_9_.

**Figure 11 nanomaterials-14-00332-f011:**
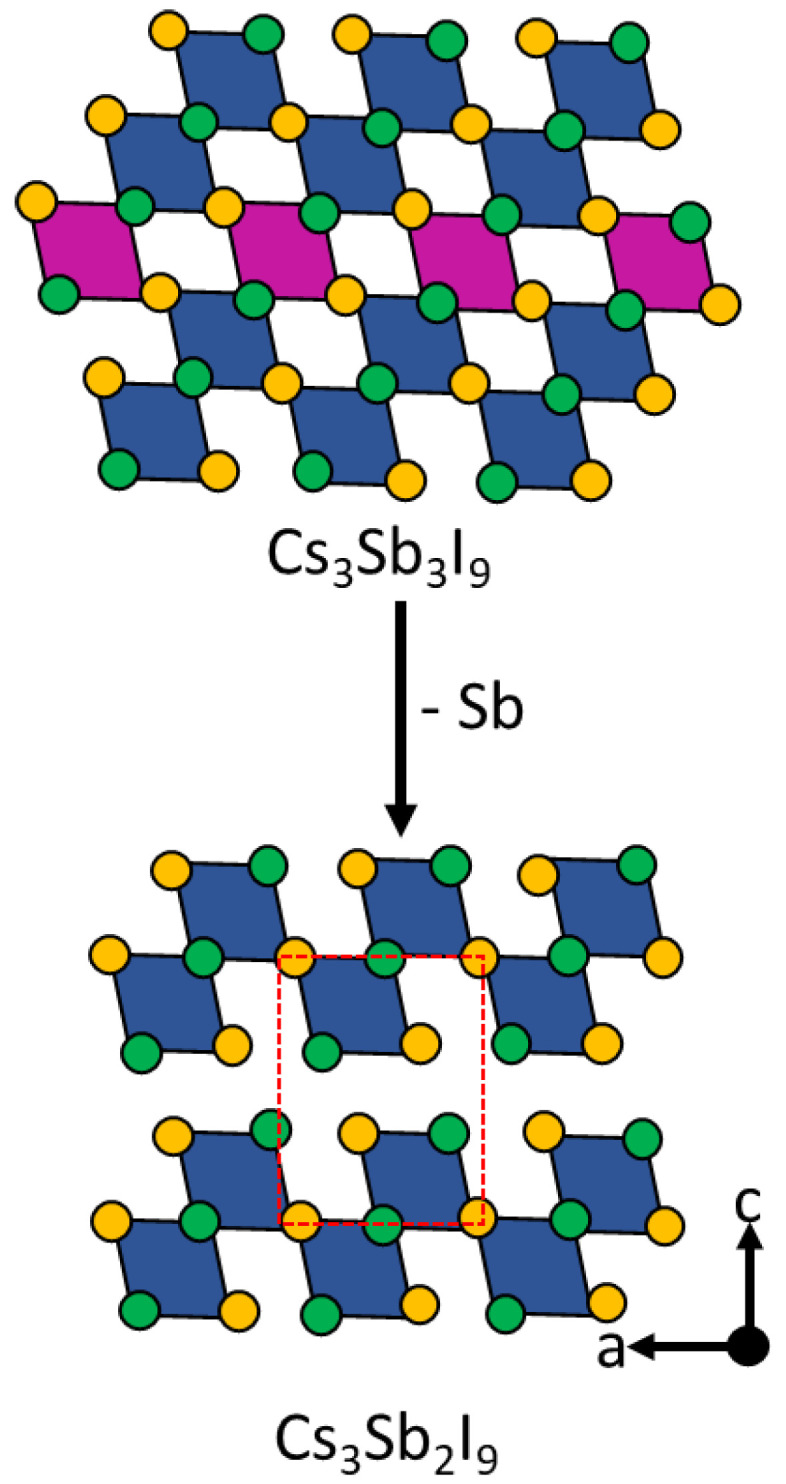
Representation of obtention of perovskite Cs_3_Sb_2_I_9_ in 2D layers (**bottom**) from the removal of the Sb layer along the ⟨111⟩ direction of the structure of perovskite Cs_3_Sb_3_I_9_ (**top**). In the figure, the orange and green spheres represent the atoms of Cs and I, respectively; Sb coordination polyhedra are shown in blue and pink. Adapted from [134].

**Figure 12 nanomaterials-14-00332-f012:**
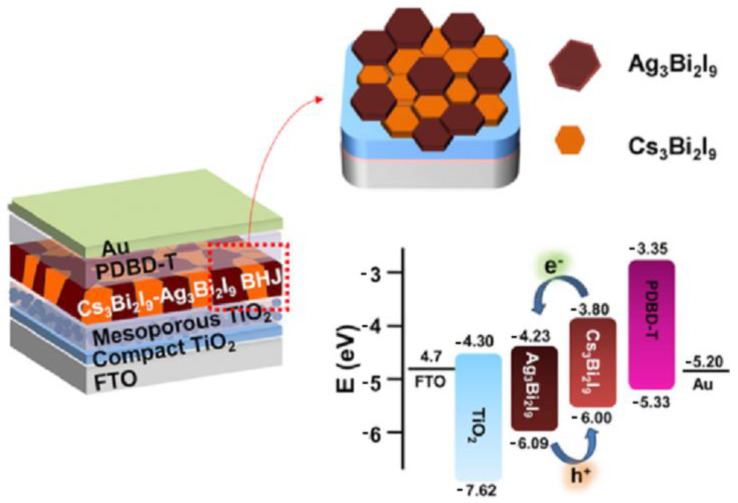
An illustrative scheme of the architecture and orientation of the Cs_3_Bi_2_I_9_ and Ag_3_Bi_2_I_9_ crystals, and an energy diagram of the cell with a heterojunction in the photoactive layer. Adapted from [150].

**Table 1 nanomaterials-14-00332-t001:** Ionic radius of some cations that can replace Pb.

Cations	Ionic Radius (Å)	Electronic Configuration
Pb^2+^	1.19	6s^2^
Sn^2+^	1.02	5s^2^
Ge^2+^	0.73	4s^2^
Bi^3+^	1.03	6s^2^
Sb^3+^	0.76	5s^2^
Sn^4+^	0.69	4d^10^
Ti^4+^	0.53	3p^6^
Cu^2+^	0.73	3d^9^

## Data Availability

Data available on request.

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
