# Peer review of "Advancements and Prospects in Perovskite Solar Cells: From Hybrid to All-Inorganic Materials"

_nanomaterials, 2024, doi:10.3390/nano14040332_

Round 1

Reviewer 1 Report

Comments and Suggestions for Authors

 This review summarizes the advancements of perovskite solar cells in the materials aspects, from hybrid to all-inorganic perovskite components, which is an important direction to solve the intrinsic stability of perovskite solar cells. Additionally, the authors analyze the materials stability from the structure to phase stability and it is the most critical items for the practical application of perovskite solar cells. However, I think the authors’ own understanding of the stability issues of perovskite solar cells should be proposed and there are still many questions should be further addressed in the manuscript. Therefore, I think that this paper should be accepted after a minor revision.

1.      There are many subscripts in the manuscript should be marked correctly such as TiO2, CH3NH3PbBr3 etc., I advise the authors check them carefully across the whole manuscript.

2.      The figure resolution is a little insufficient, please improve them, such as Figure 5.

3.      The English is a little poor, I advise the authors further polish them.

4.      In the manuscript, I advise the authors add some summarized Figures to conclude device performance (efficiency and stability progress) based on the inorganic perovskite materials to improve the readability of this review.

5.      For the reference styles, please keep them consistent with the Journal requirements. For example, some Journals are expressed by the abbreviation while some references are the full name. I advise the authors revise them carefully.

Comments on the Quality of English Language

The English is a little poor, I advise the authors further polish them.

Author Response

The authors would like to thank you for taking the time to review this manuscript. Detailed answers to the questions raised in the review can be found in the attached file.

Reviewer 2 Report

Comments and Suggestions for Authors

The manuscript provides a comprehensive review of the evolution of perovskite materials from hybrid organic/inorganic to fully inorganic compositions, as well as the partial or full substitution of lead within these structures. However, to enhance the manuscript and ensure completeness, revisions are recommended to address specific omissions.

  1. The section on tin-based perovskites overlooks the significant category of mixed tin-lead iodide perovskites. These materials are noteworthy for their smaller optical band gaps and their relevance to the field should be acknowledged and discussed.
  2. The discussion of 2D perovskites fails to reference seminal works in the field, most notably the study by Mercouri G. Kanatzidis published in Nature (2016, 536 (7616), 312–316), among others. It is crucial that these influential studies are cited to provide a thorough background on the subject. Incorporating these references and expanding the relevant sections will greatly improve the manuscript, making it a more authoritative and resourceful document for readers interested in the field of perovskite research.

Author Response

(The authors gave the same response as above.)

Reviewer 3 Report

Comments and Suggestions for Authors

The authors have presented a review paper about advancements and prospects in Perovskite Solar Cells. 

In some lines, the subindex of the compounds are not well stated, i.e.: Cs3Sb2I9. (The numbers must be in lowercase). 

Degradation mechanisms and their origins, could be explained deeper. 

Encapsulation is not assessed in the manuscript and this field is quite important for the future perspectives. 

Some perspectives have been not taken into account in the review. I suggest to enhance and remark the new promising fields, such as inks, non-polar solvents, and so on, and state them in a dedicated section. 

Author Response

(The authors gave the same response as above.)
